

# Prognostic analyses of genes associated with anoikis in breast cancer

Jingyu Cao[1,*], Xinyi Ma[2,*], Guijuan Zhang[3], Shouyi Hong[4], Ruirui Ma[4], Yanqiu Wang[4], Xianxin Yan[4] and Min Ma[4]

[1] Department of Oncology, The First Affiliated Hospital, Jinan University, Guangzhou, Guangdong, China
[2] The First Clinical Medical College, Southern Medical University, Guangzhou, China
[3] School of Nursing of Jinan University, Guangzhou, China
[4] College of Traditional Chinese Medicine, Institute of Integrated Traditional Chinese and Western Medicine, Jinan University, Guangzhou, China
[*] These authors contributed equally to this work.

## ABSTRACT

Breast cancer (BRCA) is the most diagnosed cancer worldwide and is responsible for the highest cancer-associated mortality among women. It is evident that anoikis resistance contributes to tumour cell metastasis, and this is the primary cause of treatment failure for BRCA. However, anoikis-related gene (ARG) expression profiles and their prognostic value in BRCA remain unclear. In this study, a prognostic model of ARGs based on The Cancer Genome Atlas (TCGA) database was established using a least absolute shrinkage and selection operator analysis to evaluate the prognostic value of ARGs in BRCA. The risk factor graph demonstrated that the low-risk group had longer survival than the high-risk group, implying that the prognostic model had a good performance. We identified 11 ARGs that exhibited differential expression between the two risk groups in TCGA and Gene Expression Omnibus databases. Through Gene Ontology and Kyoto Encyclopaedia of Genes and Genomes enrichment analyses, we revealed that the screened ARGs were associated with tumour progression and metastasis. In addition, a protein–protein interaction network showed potential interactions among these ARGs. Furthermore, gene set enrichment analysis suggested that the Notch and Wnt signalling pathways were overexpressed in the high-risk group, and gene set variation analysis revealed that 38 hallmark genes differed between the two groups. Moreover, Kaplan–Meier survival curves and receiver operating characteristic curves were used to identify five ARGs (CD24, KRT15, MIA, NDRG1, TP63), and quantitative polymerase chain reaction was employed to assess the differential expression of these ARGs. Univariate and multivariate Cox regression analyses were then performed for the key ARGs, with the best prediction of 3 year survival. In conclusion, ARGs might play a crucial role in tumour progression and serve as indicators of prognosis in BRCA.

Corresponding authors
Xianxin Yan, yanxianxin@jnu.edu.cn
Min Ma, tmamin@jnu.edu.cn

## INTRODUCTION

Breast cancer is the most commonly diagnosed cancer worldwide and has the highest mortality rate among malignant tumours in women (*Sung et al., 2021*). Despite comprehensive treatment, such as surgery, cytotoxic chemotherapy, and targeted

therapy, improves clinical remission rates, dissemination and metastasis often lead to the failure of anticancer therapy (*Loibl et al., 2021*). Present reports suggest that metastasis is responsible for more than 90% of breast cancer-related deaths (*Gupta & Massagué, 2006*). Therefore, the efficacy of treatment for breast cancer is largely dependent on its capacity to prevent metastasis. Tumour cell metastasis involves multiple steps, such as basement membrane invasion, intravasation and transport in the vasculature, and distant organ colonisation (*Valastyan & Weinberg, 2011*). It has been confirmed that tumour cells must overcome various stresses, including anoikis, to achieve distant metastasis (*Fanfone et al., 2022*).

Anoikis is a form of apoptosis initiated when cells detach from the extracellular matrix (ECM) and has been suggested to be regulated *via* the extrinsic death receptor pathway and intrinsic mitochondrial pathway (*Gilmore, 2005*). Usually, epithelial and endothelial cells maintain contact-dependent growth by activating the cellular signalling pathway; however, their apoptotic program is initiated after the cell–matrix contact loss, which helps maintain tissue homeostasis by eliminating shed cells (*Gilmore, 2005*). Resistance to anoikis has been reported in malignancies, including breast cancer, and is considered a fundamental characteristic of metastatic cancer cells. Therefore, anoikis is a potential therapeutic target for inhibiting cancer metastasis (*Adeshakin et al., 2021*).

Hence, anoikis has recently become the focus of intense research in the field of tumour invasion, and potential molecular mechanisms of anoikis resistance have been explored in several tumours (*Wang et al., 2022*). A previous study indicated that 14-3-3$\sigma$ induced anoikis resistance and hepatocellular carcinoma cell metastasis by epidermal growth factor receptor-dependent extracellular signal-regulated kinase 1/2 pathway activation (*Song et al., 2021*). In addition, anoikis-resistant gastric cancer cells exhibited stronger metastatic and proangiogenic traits based on CCAAT-enhancer-binding protein $\beta$-mediated platelet-derived growth factor subunit B autocrine and paracrine signalling (*Du et al., 2021*). Furthermore, anoikis-resistant molecular mechanisms have been explored in breast cancer (*Tajbakhsh et al., 2019*). Human epidermal growth factor receptor (HER) 2 and Src-dependent lactate dehydrogenase A activation contributed to breast cancer cell anoikis resistance and metastasis (*Jin et al., 2017*), and HER2-dependent B cell linker down-regulation facilitated three-dimensional breast tumour growth (*Liu et al., 2022b*). Although research has demonstrated that several genes are involved in anoikis resistance in breast cancer (*Tajbakhsh et al., 2019*), research on the clinical relevance of ARGs and the systematic evaluation of the implications of anoikis in breast cancer is lacking.

In this research, we first constructed an ARG prognostic model for breast cancer based on the Cancer Genome Atlas (TCGA) Breast Invasive Carcinoma (BRCA) (TCGA-BRCA) database and subsequently screened 11 differentially expressed ARGs between high- and low-risk groups. Functional annotation enrichment and protein-protein interaction (PPI) network analysis were performed for the screened ARGs. We identified five key ARGs by assessing their potential diagnostic and prognostic values and determined the messenger ribonucleic acid (mRNA) levels of the five key ARGs by quantitative polymerase chain reaction (qPCR). Univariate and multivariate Cox regression analyses were employed to assess the ability of 2-, 3-, and 4-year survival predictions with respect to key ARGs.

Altogether, our findings demonstrate the significant roles of ARGs in breast cancer progression and provide novel research perspectives for anoikis resistance.

## MATERIALS & METHODS

### Data collection

RNA sequencing data of patients with breast cancer were downloaded from TCGA database (https://portal.gdc.cancer.gov/) using the R package TCGAbiolinks (*Colaprico et al., 2016*). After eliminating samples with incomplete clinical annotations, 1,083 cases of breast cancer projects, including 111 cases with matched adjacent tissues, were obtained (TCGA-BRCA), and the data format was level 3 HTSeq-fragments per kilobase per million. Additionally, corresponding clinical information was obtained from the UCSC Xena database (*Goldman et al., 2020*). The mRNA counts of TCGA-BRCA dataset were normalised by using the limma R package (*Ritchie et al., 2015*).

The breast cancer-related datasets GSE42568 (*Clarke et al., 2013*), GSE102484 (*Cheng et al., 2017*), and GSE20685 (*Kao et al., 2011*) were downloaded from the Gene Expression Omnibus (GEO) database (http://www.ncbi.nlm.nih.gov/geo/) (*Barrett et al., 2013*) using the R package GEO query (*Davis & Meltzer, 2007*). The detection platform for the three datasets was GPL570 (HG-U133_Plus_2). GSE42568 included 104 breast cancer samples and 17 matching normal tissues, GSE102484 comprised 683 breast cancer samples, and GSE20685 comprised 327 breast cancer samples.

Eight gene sets of ARGs were retrieved from the Molecular Signatures Database (MSigDB) (*Liberzon et al., 2015*), which provides the most comprehensive information on human gene sets, resulting in 862 ARGs. Furthermore, 794 ARGs were retrieved from the GeneCards database (*Stelzer et al., 2016*). By taking the intersection of the two databases and omitting those genes with missing probes, we obtained 100 ARGs that are common genes in both databases (Table S1).

### Prognostic model construction

The least absolute shrinkage and selection operator (LASSO) prognostic model of 100 ARGs for breast cancer was constructed using the TCGA-BRCA dataset, employing 10-fold cross-validation with seeds 2021 and a *p*-value of 0.05. We ran 1000 iterations to guard against overfitting. The outcomes of the LASSO analysis were visualised. All samples from the TCGA-BRCA dataset and the BRCA dataset were divided into high- and low-risk groups, respectively, according to the median risk score of the prognostic model, and the survival outcomes were displayed in spot graphs. Moreover, the ARG expression profiles for prognostic model construction were visualised using heatmaps.

### Differential expression genes in the high- and low-risk groups

First, the batch effects of GSE42568, GSE20685, and GSE102484 were eliminated using the R package sva (*Leek et al., 2012*), and merged BRCA datasets were retrieved from the GEO database. Second, datasets from TCGA and GEO were standardised and then grouped into high- and low-risk groups according to the median risk score of the LASSO prognostic model. Differential gene expression analysis was performed between the two groups, and
a volcano plot (ggplot2) and heatmap (R package pheatmap) were created to visualise up-regulated genes (log FC > 0.5 and $p < 0.05$) and down-regulated genes (log FC < −0.5 and $p < 0.05$).

## Gene enrichment analysis and the PPI network

The clusterProfiler R package was used to perform Gene Ontology (GO) (*Gene Ontology Consortium, 2015*) and Kyoto Encyclopaedia of Genes and Genomes (KEGG) (*Kanehisa & Goto, 2000*) pathway enrichment analyses (*Yu et al., 2012*) with $p < 0.05$ and a false discovery rate (FDR) value ( $q$-value) of $< 0.2$. The Search Tool for the Retrieval of Interacting Genes/Proteins (STRING) database was accessed to provide information regarding known and predicted PPIs (*Szklarczyk et al., 2019*), From which information regarding ARGs was extracted from the STRING database, and a PPI network was constructed using Cytoscape (*Shannon et al., 2003*). Additionally, GeneMANIA was used for gene prediction of the selected ARGs, and the results were visualised (*Franz et al., 2018*).

## Gene set enrichment analysis (GSEA) and gene set variation analysis (GSVA)

GSEA is an analytical method that determines whether predefined gene sets demonstrate statistically significant and consistent differences between two biological states (*Subramanian et al., 2005*). In the current study, TCGA-BRCA genes were divided into phenotype-related high- and low-risk groups based on the risk score median of the LASSO prognostic model, and GSEA was performed using the clusterProfiler R package. The procedure was repeated 1,000 times for each analysis, and c2.cp.v7.2.symbols were obtained from MSigDB to serve as the reference gene collection. Statistical significance was set at FDR $< 0.25$ and $p < 0.05$.

We downloaded "h.all.v7.4.symbols.gmt" from the MSigDB database to perform GSVA based on gene expression levels in the TCGA-BRCA dataset and analyse variations of functional enrichment between the high- and low-risk groups. Statistical significance was set at $p < 0.05$.

## Analysis of the prognostic value and receiver operating characteristic (ROC) curve for ARGs

Kaplan–Meier curves were used to estimate the overall survival of patients with breast cancer based on the ARG expression derived from the TCGA-BRCA dataset. The ROC curve was used to assess the relationship between ARGs and tumourigenesis, and genes with an area under the curve (AUC) of the ROC curve $>0.6$ were considered key genes.

## Correlation analysis between clinical characteristics and prognosis

First, we performed univariate Cox regression to evaluate the prognostic value of clinical characteristics and key ARGs in patients, and a multivariate Cox regression model was constructed from factors with a $p$-value of $<0.1$ in univariate Cox regression. Subsequently, nomograms were used to predict the 2-, 3-, and 4-year survival probabilities individually based on a multivariate Cox regression model. The calibration curves were evaluated graphically by plotting the nomogram-predicted probabilities against the observed

occurrences, and the 45° line represented the best predictive values. The nomograms and calibration curves were produced by the R package rms. Decision curve analysis (DCA) was performed using the R package ggDCA (*Tataranni & Piccoli, 2019*) to evaluate the predictive effect of the nomograms for survival probability.

## Cell cultures and real-time qPCR

The human normal breast epithelial cell line MCF-10A and breast cancer cell line HCC1954, MDA-MB-231, and MCF-7 were procured from the American Type Culture Collection. MDA-MB-231 is oestrogen (ER)/progesterone (PR)/HER2 negative, HCC1954 is ER/PR negative and HER2 positive, and MCF-7 is ER/PR positive and HER2 negative. MCF-10A cells were cultured in Dulbecco's Modified Eagle Medium (DMEM)/F12 medium (GIBCO, USA) with 5% horse serum (HyClone, USA) and other supplements. MDA-MB-231and MCF-7 cells were cultured in DMEM (GIBCO, USA) containing 10% foetal bovine serum (Gibco, Billings, MT, USA), and HCC1954 cells were cultured in Rosewell Park Memorial Institute 1640 medium (GIBCO, USA) containing 10% foetal bovine serum (Gibco). All cell lines were incubated at 37 °C under 5% carbon dioxide according to the recommended protocols.

Total RNA was extracted from cells which had been cultured in dishes using TRIzol (Life Technologies, USA). Reverse transcription was then performed using an Evo M-MLV RT Mix Tracking Kit with genomic deoxyribonucleic acid (DNA) Clean (Accurate Biology, Changsha, China), and qPCR was performed using an HS SYBR Green Premix Pro Taq qPCR Tracking Kit (Accurate Biology, Changsha, China). All steps were performed according to the manufacturer's instructions. The primer sequences were as follows:

| Genes | Forward primer | Reverse primer |
| --- | --- | --- |
| CD24 | CCCCAAATCCAACTAATGC | GGACTTCCAGACGCCATT |
| Keratin 15 (KRT15) | GACGGAGATCACAGACCTGAG | CTCCAGCCGTGTCTTTATGTC |
| Melanoma-inhibiting activity (MIA) | CCTCCGTGTCCACTAAAT | TTCTTGCCGTTCATACTC |
| N-Myc downstream regulated 1 (NDRG1) | ACACCTACCGCCAGCACA | GCCACAGTCCGCCATCTT |
| Tumour protein p63 (TP63) | CCTTACATCCAGCGTTTC | TTTGTCGCACCATCTTCT |

The $\Delta\Delta Ct$ method was used to analyse qPCR data. Normalised $\Delta Ct$ values were obtained by subtracting the Ct values of the housekeeping reference gene ($\beta$-actin) from the target gene Ct values. $\Delta\Delta Ct$ values were calculated by subtracting the testing group target gene $\Delta Ct$ values from the average $\Delta Ct$ of the control group. Finally, relative mRNA expression was obtained by $2^{-\Delta\Delta Ct}$.

## Immunohistochemical (IHC) analysis

IHC information on key genes in breast cancer and normal breast tissues was obtained from the Human Protein Atlas (HPA) database (*Colwill & Gräslund, 2011*).

## Statistical analysis

Clinical data handling in the study was performed using the statistical software R (version 4.1.2; *R Core Team, 2021*). Continuous variables were summarised using the mean ± standard deviation. The Wilcoxon-rank sum test was used to compare the two groups, and Student's *t*-test was used to compare the normally distributed data of the two groups. The Kruskal–Wallis test was used for comparing more than two groups, and categorical variables were compared using the chi-square or Fisher's exact tests. Survival analyses and univariate and multivariate Cox regressions were performed using the survival package in R, and LASSO analyses were performed using the R package glmnet. Furthermore, survivalROC in the R package was used for creating the ROC curve. The significance of survival difference was tested with the log-rank test. In addition, two-sided Spearman's rank correlation coefficients were employed. One-way analysis of variance was used for comparing multiple samples of experimental data. Differences with *p*-values <0.05 were considered significant.

## RESULTS

### LASSO prognostic models were established and 11 ARGs with possible prognostic values were screened

A flow chart of the study is provided in the Fig. 1. BRCA datasets of GEO were obtained from GSE42568, GSE20685, and GSE102484 after eliminating batch effects. Box plots and principal component analysis plots displayed the datasets before and after eliminating batch effects (Figs. S1A–S1D). The results revealed that the batch effect had been effectively eliminated. We performed LASSO analysis of ARGs based on the TCGA-BRCA dataset (Fig. 2A) to evaluate the prognostic value of 100 ARGs from the intersection of MSigDB and the GeneCards database (Table S1), and the variable trajectories graph was used to visualise the results (Fig. 2B). The risk factor graph demonstrated that those in low-risk groups had longer survival compared with those in high-risk groups (Fig. 2C) and 12 ARGs (BIRC3, CD24, FGFR1, IVL, KRT15, L1CAM, MIA, NDRG1, NOS2, PTPN3, SERPINA1, and TP63) of potential prognostic value were screened (Fig. 2C) (Table 1). Next, we explored the 12 ARG profiles between two risk groups in the TCGA-BRCA and BRCA datasets, respectively, and visualised the results with pheatmaps (Figs. 2F and 2G). The box plot further showed a comparison of each ARG between the two risk groups, and there were 11 significant differentially expressed ARGs (BIRC3, CD24, IVL, KRT15, L1CAM, MIA, NDRG1, NOS2, PTPN3, SERPINA1, and TP63) with the same expression trends in both datasets (Figs. 2H and 2I).

Subsequently, we found 878 differentially expressed genes between the two risk groups based on the TCGA-BRCA dataset following thresholds of |log FC| > 0.5 and *P*.adj < 0.05. Of these, 244 genes were up-regulated, and 634 were down-regulated in the high-risk group, and the volcano plot displayed the results (Fig. 2D). In addition, 196 genes were obtained from the BRCA dataset under the same thresholds comprising 57 up-regulated genes and 139 down-regulated genes (Fig. 2E).

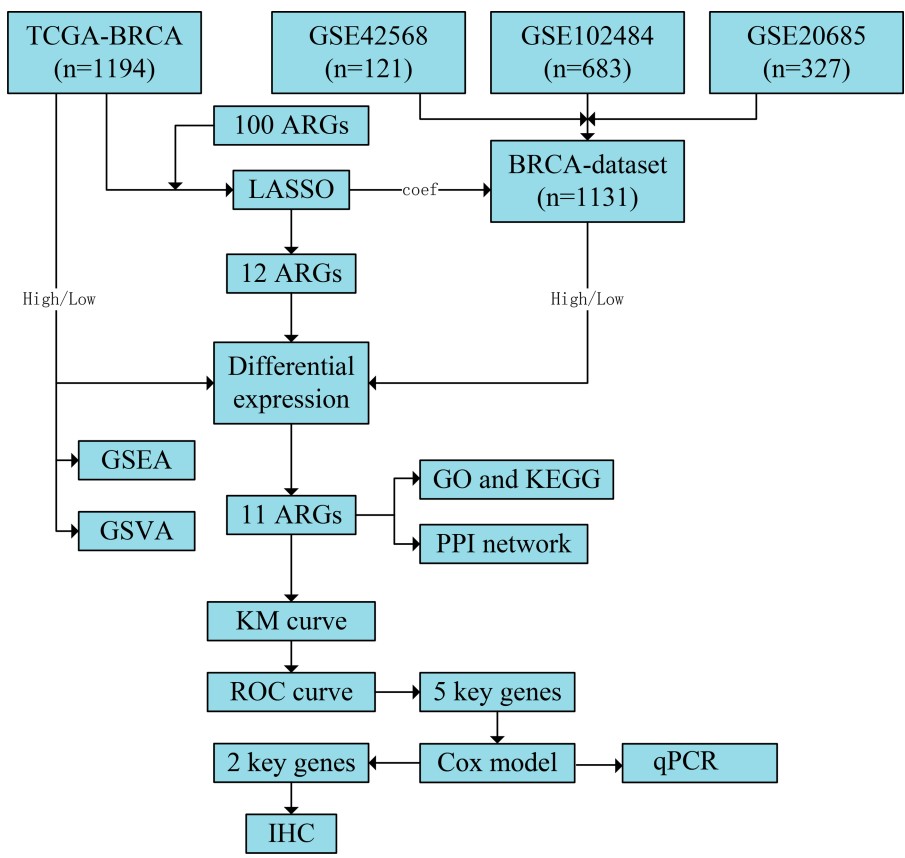

**Figure 1 Flowchart of the study.** The TCGA-BRCA and BRCA datasets were obtained from the TCGA and GEO databases, respectively. LASSO analysis was constructed using 100 ARGs based on the TCGA-BRCA dataset, and we screened 12 ARGs with possible prognostic values. Subsequently, all samples from TCGA-BRCA and BRCA datasets were divided into high- and low-risk groups based on the prognostic model's median risk score. Each ARG was compared between the two risk groups, and 11 significant differentially expressed ARGs with the same expression trends in both datasets were identified. Next, GO and KEGG enrichment analyses and PPI network construction were performed using the 11 ARGs. We further performed GSEA and GSVA based on the TCGA-BRCA dataset and obtained five key ARGs by plotting KM survival curves and ROC curves (AUC > 0.6) for each ARG based on the TCGA-BRCA dataset. Finally, univariate and multivariate Cox regression analyses were performed for key ARGs, and qPCR was used to detect gene expression in different cell lines. Results of ARG protein expression with an HR of >1 were downloaded from the HPA database. TCGA, The Cancer Genome Atlas; BRCA, breast invasive carcinoma; GEO, Gene Expression Omnibus; LASSO, least absolute shrinkage and selection operator; ARGs, anoikis-related genes; GO, Gene Ontology; KEGG, Kyoto Encyclopaedia of Genes and Genomes; PPI, protein-protein interaction; GSEA, gene set enrichment analysis; GSVA, gene set variation analysis; KM, Kaplan–Meier curve; ROC, receiver operating characteristic curve; AUC, area under the curve; qPCR, quantitative polymerase chain reaction; HR, hazard ratio; HPA, human protein atlas.

## Results of the gene enrichment analysis and PPI network based on screened ARGs

We performed GO and KEGG enrichment analysis, including cellular components, biological processes (BPs), molecular functions (MFs), and signalling pathways, with the criteria of $p < 0.05$ and FDR ($q$-value) $< 0.25$ (Table 2) to elucidate the biological functions of the selected 11 ARGs. GO analysis demonstrated that 11 ARGs are predominantly

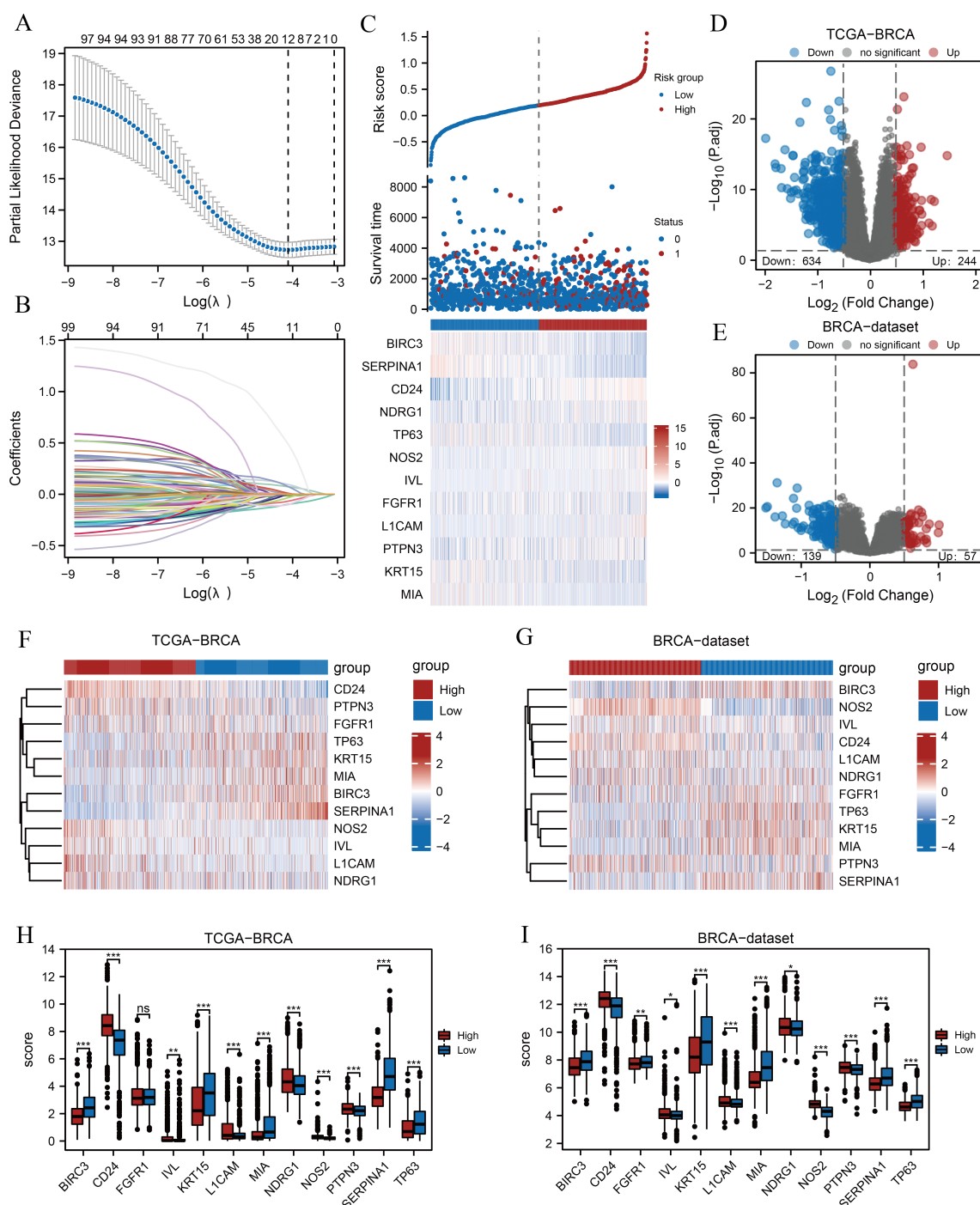

**Figure 2** **Prognostic model establishment and DEG screening between the high- and low-risk groups in breast cancer.** (A) A LASSO prognostic model was constructed using 100 ARGs. (B–C) Variable trajectories graph and risk factor graph of the prognostic model. (D–E) Volcanic maps of the DEGs between the two risk groups based on the TCGA-BRCA and BRCA datasets, respectively. (F–G) Heat maps of ARG expression between the two risk groups in TCGA-BRCA and BRCA datasets, respectively. (H–I) Box plots comparing each ARG between the two risk groups in TCGA-BRCA and BRCA datasets. ns, $p \geq 0.05$, *, $p < 0.05$, **, $p < 0.01$, ***, $p < 0.001$, and $p < 0.05$ were considered statistically significant. DEGs, differentially expressed genes; LASSO, least absolute shrinkage and selection operator; TCGA, The Cancer Genome Atlas; BRCA, breast invasive carcinoma; ARGs, anoikis-related genes.

**Table 1  List of description and expression difference of anoikis-related genes of the LASSO model.**

| Gene symbol | Description | log FC | *P*.Value | adj. *P* |
|---|---|---|---|---|
| BIRC3 | Baculoviral IAP Repeat Containing 3 | −0.551334371 | 1.89883E−10 | 4.32641E−09 |
| CD24 | CD24 Molecule | 0.406051661 | 0.000275766 | 0.001024153 |
| FGFR1 | Fibroblast Growth Factor Receptor 1 | 0.19873266 | 0.008331891 | 0.01969228 |
| IVL | Involucrin | 0.606150008 | 0.000322142 | 0.001169971 |
| KRT15 | Keratin 15 | −1.257950408 | 3.6222E−15 | 3.52105E−13 |
| L1CAM | L1 Cell Adhesion Molecule | 0.319550324 | 0.019858044 | 0.041563644 |
| MIA | MIA SH3 Domain Containing | −1.265363973 | 5.8105E−14 | 4.00434E−12 |
| NDRG1 | N-Myc Downstream Regulated 1 | 0.371969381 | 2.8489E−06 | 1.86671E−05 |
| NOS2 | Nitric Oxide Synthase 2 | 0.482118761 | 4.97052E−12 | 1.8252E−10 |
| PTPN3 | Protein Tyrosine Phosphatase Non-Receptor Type 3 | 0.034313831 | 0.482891244 | 0.581437763 |
| SERPINA1 | Serpin Family A Member 1 | −0.677585034 | 4.08221E−09 | 6.07581E−08 |
| TP63 | Tumor Protein P63 | −0.865989129 | 2.1598E−10 | 4.82553E−09 |

**Table 2  GO and KEGG enrichment analysis results of anoikis-related genes.**

| Ontology | ID | Description | Gene ratio | Bg ratio | *p* value | *p*. adjust | *q* value |
|---|---|---|---|---|---|---|---|
| BP | GO:0030216 | keratinocyte differentiation | 3/11 | 305/18670 | 6.46e−04 | 0.096 | 0.057 |
| BP | GO:0001666 | response to hypoxia | 3/11 | 359/18670 | 0.001 | 0.096 | 0.057 |
| BP | GO:0045861 | negative regulation of proteolysis | 3/11 | 363/18670 | 0.001 | 0.096 | 0.057 |
| MF | GO:0004866 | endopeptidase inhibitor activity | 2/11 | 175/17697 | 0.005 | 0.097 | 0.059 |
| MF | GO:0030414 | peptidase inhibitor activity | 2/11 | 182/17697 | 0.005 | 0.097 | 0.059 |
| MF | GO:0010181 | FMN binding | 1/11 | 16/17697 | 0.010 | 0.099 | 0.061 |
| KEGG | hsa05222 | Small cell lung cancer | 2/7 | 92/8076 | 0.003 | 0.071 | 0.064 |
| KEGG | hsa05145 | Toxoplasmosis | 2/7 | 112/8076 | 0.004 | 0.071 | 0.064 |

**Notes.**

GO, Gene Ontology; BP, biological process; CC, cellular component; MF, molecular function; KEGG, Kyoto Encyclopedia of Genes and Genomes.

enriched in BP-related processes (such as keratinocyte differentiation, response to hypoxia, and negative proteolysis regulation) and MF-related processes (such as endopeptidase inhibitor activity, peptidase inhibitor activity, and flavin mononucleotide [FMN] binding) (Fig. S2A). KEGG pathway analysis showed that the 11 ARGs were primarily associated with small-cell lung cancer and toxoplasmosis (Fig. S2A). Figure S2B illustrates the potential links between these items. Furthermore, the chord diagram illustrated each ARG's distribution in the GO items and KEGG pathways (Fig. S2C). CD24, NOS2, and NDRG1 were associated with the process of response to hypoxia. KRT15, TP63, and IVL were mainly enriched in the process of keratinocyte differentiation. BIRC3 and NOS2 were closely associated with small-cell lung cancer pathways. SERPINA1 and BIRC3 were associated with peptidase inhibitor activity.

PPI networks were constructed using the STRING database with an interaction threshold of 0.15 to ascertain the interactions between the 11ARGs in breast cancer. Visualisations were performed using the Cytoscape software (Fig. S2D). The results indicated potential interaction between ARGs. Furthermore, genes with similar functions to ARGs were predicted, and a protein network was constructed using GeneMANIA, with ephrin type-B
receptor 2, RAN binding protein 9, and activated leukocyte cell adhesion molecule being closely associated with the 11 ARGs (Fig. S2E).

## Signalling pathways and hallmark gene sets associated with risk groupings

We subsequently performed GSEA and GSVA between high- and low-risk groups based on the TCGA-BRCA dataset to explore potential relationships between ARGs and gene sets related to breast cancer. The results of GSEA indicated several significantly enriched BPs (Table 3), including pre-Notch expression and processing (Fig. 3B), signalling by Notch (Fig. 3C), T cell factor-dependent signalling in response to Wnt (Fig. 3D), and signalling by Wnt (Fig. 3E), as illustrated in Fig. 3A. Other BPs, including transcription of androgen receptor-regulated genes kallikrein-related peptidase (KLK) 2 and KLK3, DNA damage telomere stress-induced senescence, and processes associated with cell cycle regulation, are presented in Table 3. We performed GSVA on all genes in TCGA-BRCA to further explore the differentially expressed hallmark gene sets between the two risk groups. Table 4 reveals that 38 hallmarks exhibited differences, which were visualised using heat maps (Fig. 3F), and the gene sets with $p$-values $<0.001$ were compared between two risk groups and visualised using box plot diagrams (Fig. 3G). The results implied that gene sets of Myc targets, E2F targets, hypoxia, p53, tumour necrosis factor (TNF)-$\alpha$/nuclear factor kappa B (NF-$\kappa$B), interleukin (IL)-6/Janus kinase (JAK)/signal transducer and activator of transcription 3 (STAT3), and mammalian target of rapamycin complex 1 (mTORC1) signalling pathway had marked differences in their enrichment between the two groups.

## Analysis of the prognostic significance and ROC curve of the ARGs

Kaplan–Meier survival curves were plotted for 11 ARGs based on the TCGA-BRCA dataset (Figs. 4A–4I) to further explore the roles of ARGs in the prognosis of patients with breast cancer. We found that nine ARGs impacted the prognosis of patients with breast cancer ($p < 0.05$). Specifically, BIRC3 (hazard ratio [HR] $= 0.67$, $p = 0.015$), KRT15 (HR $= 0.61$, $p = 0.002$), MIA (HR $= 0.68$, $p = 0.021$), SERPINA1 (HR $= 0.58$, $p = 0.001$), and TP63 (HR $= 0.64$, $P = 0.007$) were positively correlated with overall survival, while CD24 (HR $= 1.69$, $p = 0.002$), IVL (HR $= 1.71$, $p = 0.001$), NDRG1 (HR $= 1.41$, $p = 0.036$), and NOS2 (HR $= 1.46$, $p = 0.022$) were associated with a poor prognosis. We also performed ROC curves to evaluate the correlation between ARGs and tumourigenesis using TCGA-BRCA dataset. As a result, the AUC of the ROC curve was calculated, and five key ARGs (CD24, KRT15, MIA, NDRG1, and TP63) with AUC $> 0.6$ were demonstrated (Figs. 4J–4N).

## Correlation analysis between the clinical characteristics and prognosis

Clinical information of patients from TCGA-BRCA dataset was extracted to assess the LASSO prognostic model (Table 5). We initially performed univariate Cox regression to evaluate the prognostic value of crucial ARGs (CD24, KRT15, MIA, NDRG1, and TP63) in patients and then formulated a multivariate Cox regression model comprising of those factors with a $p$-value $< 0.1$ in the univariate Cox regression (Table 6). Forest plots revealed

**Table 3  GSEA analysis of the TCGA-BRCA dataset.**

| Description | Set size | Enrichment score | NES | *p* value | *p*. adjust |
|---|---|---|---|---|---|
| Condensation of prophase chromosomes | 72 | 0.648142653 | 2.609133444 | 0.000384468 | 0.007527598 |
| DNA methylation | 63 | 0.64924649 | 2.550957964 | 0.000368053 | 0.007527598 |
| Deposition of new cenpa containing nucleosomes at the centromere | 73 | 0.62272815 | 2.511036721 | 0.000387747 | 0.007527598 |
| Meiotic recombination | 85 | 0.597551004 | 2.469681726 | 0.000414594 | 0.007527598 |
| SIRT1 negatively regulates rrna expression | 66 | 0.615267088 | 2.433992405 | 0.000374532 | 0.007527598 |
| PRC2 methylates histones and DNA | 71 | 0.599962935 | 2.413657315 | 0.000381825 | 0.007527598 |
| Activated PKN1 stimulates transcription of AR androgen receptor regulated genes KLK2 and KLK3 | 65 | 0.598544909 | 2.36548707 | 0.000370508 | 0.007527598 |
| Recognition and association of DNA glycosylase with site containing an affected purine | 56 | 0.603186125 | 2.318484147 | 0.000351 | 0.007527598 |
| DNA damage telomere stress induced senescence | 80 | 0.563392337 | 2.306378423 | 0.000403551 | 0.007527598 |
| Mitotic prophase | 141 | 0.504722456 | 2.267243796 | 0.000542005 | 0.008140373 |
| Nonhomologous end joining NHEJ | 69 | 0.566496673 | 2.266201035 | 0.000376223 | 0.007527598 |
| Pre NOTCH expression and processing | 107 | 0.467738962 | 2.020954761 | 0.000458295 | 0.007527598 |
| Signaling by NOTCH | 234 | 0.287501505 | 1.378143085 | 0.003220612 | 0.03396672 |
| TCF dependent signaling in response to WNT | 232 | 0.286479654 | 1.373337984 | 0.006405124 | 0.058235094 |
| Signaling by WNT | 329 | 0.262444276 | 1.302165188 | 0.006944444 | 0.060142944 |

**Notes.**

GSEA, Gene Set Enrichment Analysis; TCGA, The Cancer Genome Atlas; BRCA, breast invasive carcinoma.

that two genes (CD24 and NDRG1) were associated with increased risk with HRs > 1, while three other genes (KRT15, MIA, and TP63) were protective genes with HRs < 1 (Fig. 5A). Subsequently, we constructed nomograms for ARGs to evaluate the predictive power of the Cox regression models (Fig. 5B). Furthermore, we performed calibration curves for 2-, 3-, and 4-year survival predictive power of Cox regression model nomograms, respectively (Figs. 5C–5E). The results indicated that the blue line corresponding to the 3-year survival was closest to the ideal diagonal 45° line (grey line), implying that the Cox regression models yielded the most reliable prediction for the 3-year survival. Finally, DCA was used to assess the clinical applicability of the LASSO-Cox prognostic model for the 2-, 3-, and 4-year survival (Figs. 5F–5H), and the blue line in the graphs represents the model's predictive power. The range on the *x*-axis of the blue line was higher than that of the red line (all positive) and grey line (all negative) for the 3-year survival, indicating the best clinical applicability in terms of the 3-year survival.

## mRNA levels of ARGs in MCF-10A, MDA-MB-231, HCC1954, and MCF-7 cell lines

Gene mRNA levels of key ARGs in normal mammary and breast cancer cell lines were detected using qPCR. We ascertained that CD24 expression was up-regulated in breast cancer cells compared with normal breast epithelial cells (Fig. 6A). Moreover, KRT15, MIA, and TP63 levels significantly decreased in breast cancer cells (Figs. 6B, 6C and 6E). However, mRNA levels of NDRG1 varied across the three breast cancer lines, being

**Table 4  GSVA analysis of the TCGA-BRCA dataset.**

| Ontology | log FC | Ave Expr | t | P. value | adj. P |
|---|---|---|---|---|---|
| HALLMARK_P53_PATHWAY | 0.119329283 | −0.031254621 | 8.399741679 | 1.39E−16 | 6.95E−15 |
| HALLMARK_IL6_JAK_STAT3_SIGNALING | 0.160554553 | −0.019901999 | 7.747293348 | 2.15E−14 | 5.38E−13 |
| HALLMARK_TNFA_SIGNALING_VIA_NFKB | 0.149491843 | −0.018780161 | 7.593211543 | 6.72E−14 | 1.12E−12 |
| HALLMARK_ALLOGRAFT_REJECTION | 0.165689567 | −0.01910031 | 7.335435252 | 4.33E−13 | 5.41E−12 |
| HALLMARK_G2M_CHECKPOINT | −0.160835941 | −0.030063227 | −7.23791024 | 8.62E−13 | 8.62E−12 |
| HALLMARK_KRAS_SIGNALING_DN | 0.077950216 | 0.027929351 | 7.179824705 | 1.30E−12 | 1.08E−11 |
| HALLMARK_INTERFERON_GAMMA_RESPONSE | 0.162439186 | −0.023700528 | 6.976246774 | 5.27E−12 | 3.76E−11 |
| HALLMARK_APOPTOSIS | 0.105082386 | −0.016017823 | 6.93050969 | 7.19E−12 | 4.49E−11 |
| HALLMARK_INFLAMMATORY_RESPONSE | 0.1292374 | −0.01320064 | 6.588259013 | 6.93E−11 | 3.85E−10 |
| HALLMARK_E2F_TARGETS | −0.160184208 | −0.027553265 | −6.55175209 | 8.78E−11 | 4.39E−10 |
| HALLMARK_MITOTIC_SPINDLE | −0.107442266 | −0.026580477 | −6.479749206 | 1.39E−10 | 6.33E−10 |
| HALLMARK_IL2_STAT5_SIGNALING | 0.10427956 | −0.023708254 | 6.38643486 | 2.52E−10 | 1.05E−09 |
| HALLMARK_INTERFERON_ALPHA_RESPONSE | 0.162750622 | −0.025587788 | 6.349727433 | 3.17E−10 | 1.13E−09 |
| HALLMARK_SPERMATOGENESIS | −0.079775897 | 0.024668073 | −6.349696603 | 3.17E−10 | 1.13E−09 |
| HALLMARK_COAGULATION | 0.102292767 | 0.017160336 | 6.17528166 | 9.33E−10 | 3.11E−09 |
| HALLMARK_ESTROGEN_RESPONSE_LATE | 0.09490817 | −0.013436064 | 6.131517793 | 1.22E−09 | 3.81E−09 |
| HALLMARK_COMPLEMENT | 0.103046183 | −0.015778926 | 6.011855981 | 2.50E−09 | 7.37E−09 |
| HALLMARK_XENOBIOTIC_METABOLISM | 0.075977134 | 0.000803129 | 5.709659661 | 1.46E−08 | 4.06E−08 |
| HALLMARK_MYC_TARGETS_V1 | −0.110050823 | −0.039521363 | −5.536657383 | 3.87E−08 | 9.83E−08 |
| HALLMARK_ESTROGEN_RESPONSE_EARLY | 0.098760419 | 0.003536364 | 5.533710477 | 3.93E−08 | 9.83E−08 |
| HALLMARK_KRAS_SIGNALING_UP | 0.093897872 | −0.006226696 | 5.43040401 | 6.94E−08 | 1.65E−07 |
| HALLMARK_WNT_BETA_CATENIN_SIGNALING | 0.081765544 | −0.029949865 | 4.613361613 | 4.44E−06 | 1.01E−05 |
| HALLMARK_MTORC1_SIGNALING | −0.08067193 | −0.030607917 | −4.531864271 | 6.50E−06 | 1.41E−05 |
| HALLMARK_MYC_TARGETS_V2 | −0.094920667 | −0.035952966 | −3.947820502 | 8.40E−05 | 0.000174914 |
| HALLMARK_BILE_ACID_METABOLISM | 0.053809556 | 0.000674898 | 3.933254506 | 8.91E−05 | 0.000178277 |
| HALLMARK_PROTEIN_SECRETION | −0.068629916 | −0.00796288 | −3.623357806 | 0.000304306 | 0.000585204 |
| HALLMARK_HYPOXIA | 0.053946213 | −0.019883061 | 3.427859931 | 0.000631247 | 0.001139569 |
| HALLMARK_UNFOLDED_PROTEIN_RESPONSE | −0.053072207 | −0.042682246 | −3.424870074 | 0.000638159 | 0.001139569 |
| HALLMARK_APICAL_SURFACE | 0.05269076 | 0.000821367 | 3.364386226 | 0.00079404 | 0.001369035 |
| HALLMARK_FATTY_ACID_METABOLISM | 0.045000694 | −0.013810827 | 3.099805318 | 0.001986498 | 0.003302312 |
| HALLMARK_APICAL_JUNCTION | 0.049957108 | −0.006908797 | 3.090764305 | 0.002047433 | 0.003302312 |
| HALLMARK_ADIPOGENESIS | 0.041776924 | −0.029142429 | 2.799430327 | 0.005210284 | 0.008141069 |
| HALLMARK_MYOGENESIS | 0.044253317 | −0.00702594 | 2.695291931 | 0.007141694 | 0.010820749 |
| HALLMARK_GLYCOLYSIS | −0.038843193 | −0.042366566 | −2.663039667 | 0.007858759 | 0.011556999 |

Cao et al. (2023), *PeerJ*, DOI 10.7717/peerj.15475

**Table 4** (*continued*)

| Ontology | log FC | Ave Expr | t | P. value | adj. P |
|---|---|---|---|---|---|
| HALLMARK_NOTCH_SIGNALING | 0.047821813 | −0.018464182 | 2.570911437 | 0.010275841 | 0.014679772 |
| HALLMARK_PANCREAS_BETA_CELLS | 0.036594431 | 0.073863399 | 2.521151193 | 0.011839923 | 0.016444337 |
| HALLMARK_HEME_METABOLISM | 0.028007544 | −0.020449587 | 2.287809617 | 0.022340814 | 0.030190289 |
| HALLMARK_REACTIVE_OXYGEN_SPECIES_PATHWAY | 0.036597558 | −0.039232763 | 2.049238101 | 0.040679537 | 0.053525707 |

**Notes.**

GSVA, Gene Set Variation Analysis; TCGA, The Cancer Genome Atlas; BRCA, breast invasive carcinoma.

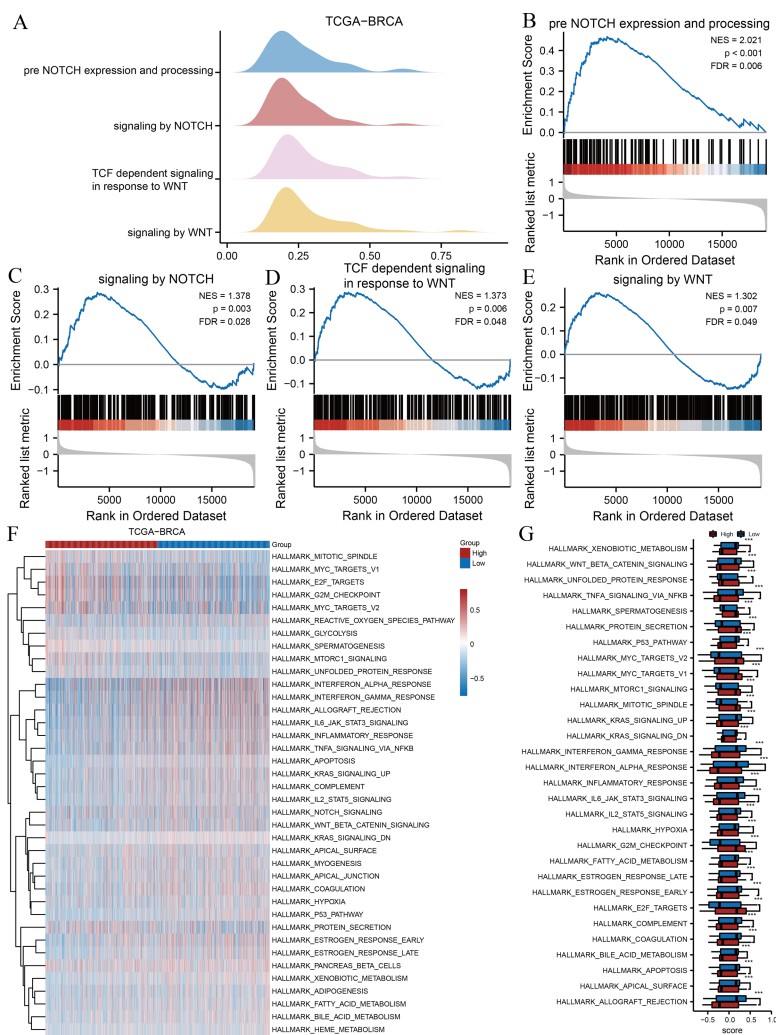

**Figure 3  GSEA and GSVA.** (A) Four biological processes of GSEA for TCGA-BRCA dataset. Enrichment analysis of the pre-Notch expression and processing (B), signalling by Notch (C), TCF dependent signalling in response to Wnt (D), signalling by Wnt (E) in TCGA-BRCA dataset. (F) Heat maps of GSVA for TCGA-BRCA dataset. (G) Box plots of significantly differently enriched hallmark genes between the two risk groups. ***, $p < 0.001$ and $p < 0.05$ were considered statistically significant. GSEA, gene set enrichment analysis; TCGA, The Cancer Genome Atlas; BRCA, breast invasive carcinoma; TCF, T cell factor; GSVA, gene set variation analysis.

up-regulated in MDA-MB-231 cells and down-regulated in HCC1954 and MCF-7 cells (Fig. 6D).

## CD24 and NDRG1 protein levels in breast cancer tissues

The expression levels of CD24 and NDRG1 that had an HR of >1 in the multivariable Cox regression model were detected using IHC based on the HPA database. It was observed, in comparison to normal breast tissues, there was a significant increase in the CD24 and NDRG1 expression in breast cancer (Figs. S3A–S3D).

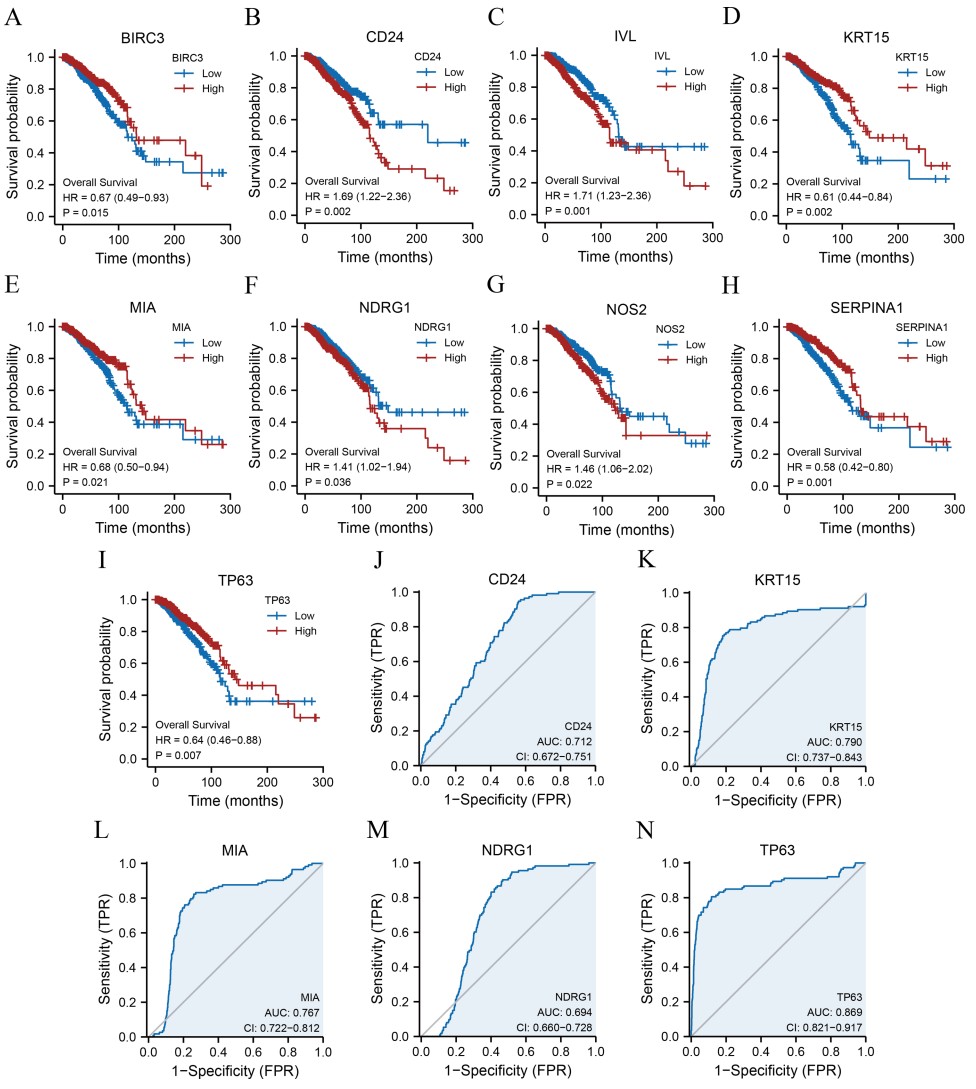

**Figure 4 KM survival curves and ROC curves.** KM curves of BIRC3 (A), CD24 (B), IVL (C), KRT15 (D), MIA (E), NDRG1 (F), NOS2 (G), SERPINA1 (H), and TP63 (I). ROC curves of CD24 (J), KRT15 (K), MIA (L), NDRG1 (M), and TP63 (N). KM, Kaplan–Meier; BIRC3, baculoviral IAP repeat-containing protein 3; CD24, cluster of differentiation 24; IVL, involucrin; MIA, melanoma-inhibiting activity; KRT15, keratin 15; NDRG1, N-Myc downstream regulated 1; NOS2, nitric oxide synthase 2; SERPINA1, serpin family A member 1; TP63, tumour protein p63; ROC, receiver operating characteristic.

# DISCUSSION

Breast cancer remains a serious threat to women due to its high incidence and mortality rates (*Sung et al., 2021*), with distant metastasis being the primary cause of breast cancer-associated mortality worldwide (*Liang et al., 2020*). However, treatments for metastasis remain elusive. Therefore, it is necessary to study the underlying molecular mechanisms of metastasis, while at the same time, there is an urgent need to explore novel clinical diagnostic and prognostic biomarkers and therapeutic targets for breast cancer. Anoikis

**Table 5  Patient characteristics of BRCA patients in the TCGA datasets.**

| Characteristic | Levels | Overall |
|---|---|---|
| *n* | | 1083 |
| T stage, *n* (%) | T1 | 277 (25.6%) |
| | T2 | 629 (58.2%) |
| | T3 | 139 (12.9%) |
| | T4 | 35 (3.2%) |
| N stage, *n* (%) | N0 | 514 (48.3%) |
| | N1 | 358 (33.6%) |
| | N2 | 116 (10.9%) |
| | N3 | 76 (7.1%) |
| M stage, *n* (%) | M0 | 902 (97.8%) |
| | M1 | 20 (2.2%) |
| Pathologic stage, *n* (%) | Stage I | 181 (17.1%) |
| | Stage II | 619 (58.4%) |
| | Stage III | 242 (22.8%) |
| | Stage IV | 18 (1.7%) |
| Age, *n* (%) | <=60 | 601 (55.5%) |
| | >60 | 482 (44.5%) |
| OS event, *n* (%) | Alive | 931 (86%) |
| | Dead | 152 (14%) |
| DSS event, *n* (%) | Alive | 978 (92%) |
| | Dead | 85 (8%) |
| PFI event, *n* (%) | Alive | 936 (86.4%) |
| | Dead | 147 (13.6%) |
| Age, median (IQR) | | 58 (48.5, 67) |

resistance has been established as a hallmark of advanced malignancy due to its ability to promote tumour cell invasion and metastasis (*Adeshakin et al., 2021*). Recently, researchers focused more on ARGs due to their potential prognostic and therapeutic value (*Liu et al., 2022a*). However, a comprehensive understanding of ARGs in relation to breast cancer is lacking.

In this study, we performed a LASSO analysis based on the TCGA-BRCA dataset through ARGs and divided all samples of TCGA-BRCA and BRCA datasets into high- and low-risk groups based on the median risk score of the prognostic model. Then we screened 11 significant differentially expressed ARGs between two risk groups with the same expression patterns in both datasets, suggesting their potential prognostic value in breast cancer samples. Thus, to further investigate the potential function and associated molecular mechanisms of the screened ARGs, we performed function and pathway enrichment analysis. The results revealed that ARGs were primarily enriched in the processes of response to hypoxia, proteolysis regulation, peptidase inhibitor activity, FMN binding, and small-cell lung cancer. To the best of our knowledge, resistance to a hypoxic environment, which is one of the hallmarks of tumour cells, could enhance the anoikis resistance of tumour cells and promote cell survival and distant metastasis (*Wang et al., 2021*; *Wobma et*

**Table 6  Cox regression to identify hub genes and clinical features associated with OS.**

| Characteristics | Total ($N$) | Univariate analysis | | Multivariate analysis | |
|---|---|---|---|---|---|
| | | Hazard ratio (95% CI) | *P* value | Hazard ratio (95% CI) | *P* value |
| CD24 | 1082 | | | | |
| Low | 540 | Reference | | | |
| High | 542 | 1.694 (1.217–2.359) | **0.002** | 1.583 (1.132–2.213) | **0.007** |
| KRT15 | 1082 | | | | |
| Low | 540 | Reference | | | |
| High | 542 | 0.606 (0.438–0.837) | **0.002** | 0.673 (0.468–0.969) | **0.033** |
| MIA | 1082 | | | | |
| Low | 540 | Reference | | | |
| High | 542 | 0.685 (0.497–0.944) | **0.021** | 0.836 (0.568–1.231) | 0.365 |
| NDRG1 | 1082 | | | | |
| Low | 540 | Reference | | | |
| High | 542 | 1.410 (1.023–1.944) | **0.036** | 1.409 (1.010–1.966) | **0.043** |
| TP63 | 1082 | | | | |
| Low | 540 | Reference | | | |
| High | 542 | 0.640 (0.464–0.883) | **0.007** | 0.798 (0.558–1.142) | 0.218 |

**Notes.**
OS, overall survival.

*al., 2018*). As predicted, our results remain consistent with those previous studies that found ARGs to be enriched in hypoxia-related functional processes. Proteolysis is a fundamental metabolic process essential for life and abnormal regulation of proteolysis and peptidase activity are associated with cancer. For example, tumour cell metastasis can be facilitated by the degradation of the extracellular matrix by secreted proteinases (*Osuala et al., 2019*). FMN, which is a cofactor of enzymes, has been the focus of photodynamic therapy for tumours (*Yang, Chang & Chen, 2017*). Therefore, our data enhances our understanding of ARGs in breast cancer and guides further research.

It is now evident that activation of the Notch and WNT signalling pathway was associated with anoikis resistance in various malignancies, including breast cancer (*Tan et al., 2019*; *Kwon et al., 2014*; *Leong et al., 2007*). GSEA results revealed that the Notch and Wnt signalling pathways were enriched in the high-risk group, which implies that the poor prognosis of patients with breast cancer in the model might be associated with abnormal pathway activation. Notably, the results further revealed potential regulatory relationships between ARGs and the Notch and Wnt signalling pathways, which need to be further explored in the future. Growing evidence has indicated that the androgen receptor modulates breast cancer progression (*Magklara, Brown & Diamandis, 2002*). However, the relationship between anoikis and the expression of androgen receptor-regulated genes KLKs remains unclear. Herein, GSEA revealed that ARGs might be involved in the process of transcription of genes KLK2 and KLK3 in breast cancer, which might offer clues for a more in-depth investigation of ARGs and anoikis resistance in breast cancer. In addition, GSVA results revealed that 30 gene sets exhibited significant differences between the two risk groups, comprising Myc targets, E2F targets, hypoxia, p53, TNF $\alpha$/NF-$\kappa$B,

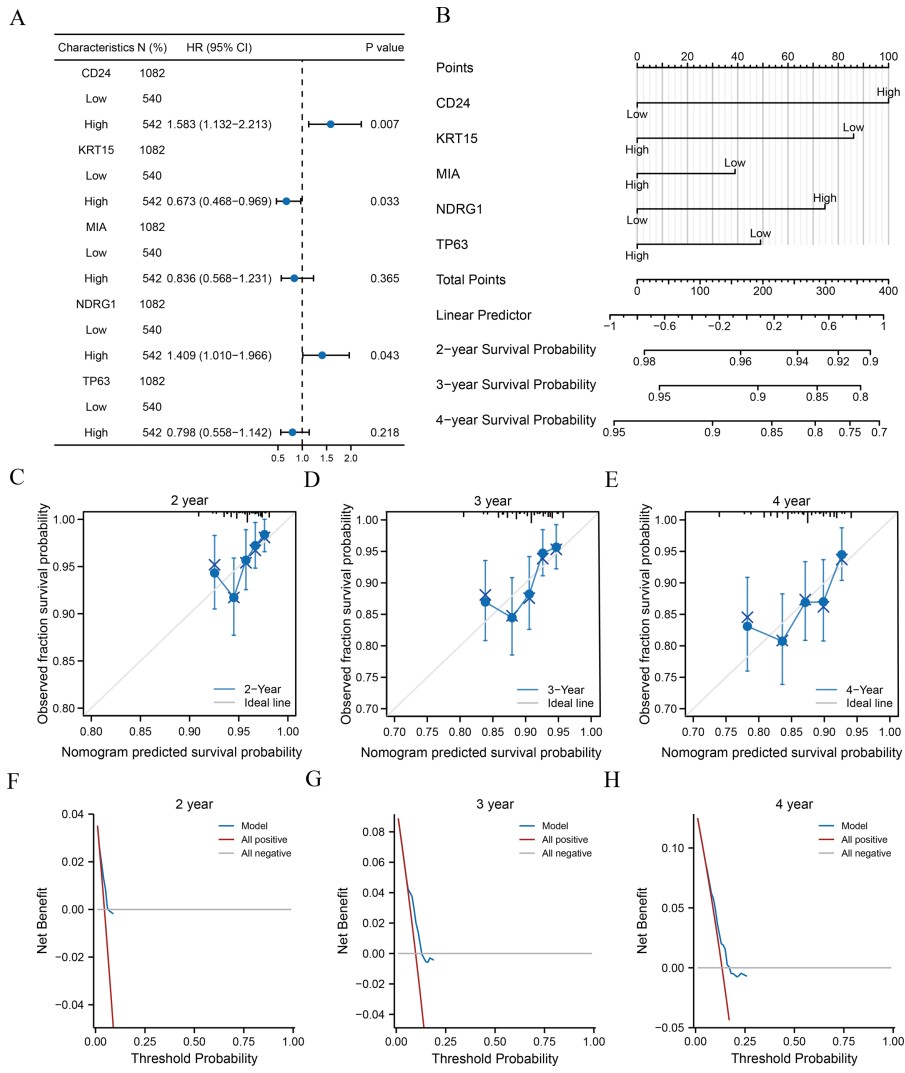

**Figure 5  Analysis of clinical correlation with the prognostic models.** (A–B) Forest plots (A) and nomograms (B) of the multivariate Cox regression model for key ARGs. (C–E) Calibration curves of the 2- (C), 3- (D), and 4-year (E) survival for nomograms of multivariate Cox regression. Black lines at the top correspond to the distributions of predicted survival probabilities of samples. (F-H) 2- (F), 3- (G), and 4-year (H) DCA of multivariate Cox regression. ARGs, anoikis-related genes; DCA, decision curve analysis.

IL-6/JAK/STAT3, and mTORC1 signalling pathways, implying a potential correlation between these pathways and the role of ARGs in the invasion and metastasis of breast cancer. Previous studies have demonstrated that p53 induced anoikis and decreased breast cancer metastasis (*Cheng et al., 2009*), and mTORC1 has been reported to be associated with anoikis in breast cancer (*Ng et al., 2012*). This further supported our study. However, the relationship between anoikis and Myc and E2F in breast cancer is yet to be identified. We uncovered their potential association in breast cancer in the study.

Subsequently, to elucidate the prognostic value of ARGs and their correlation with breast cancer progression, we plotted survival and ROC curves individually. As a result, five

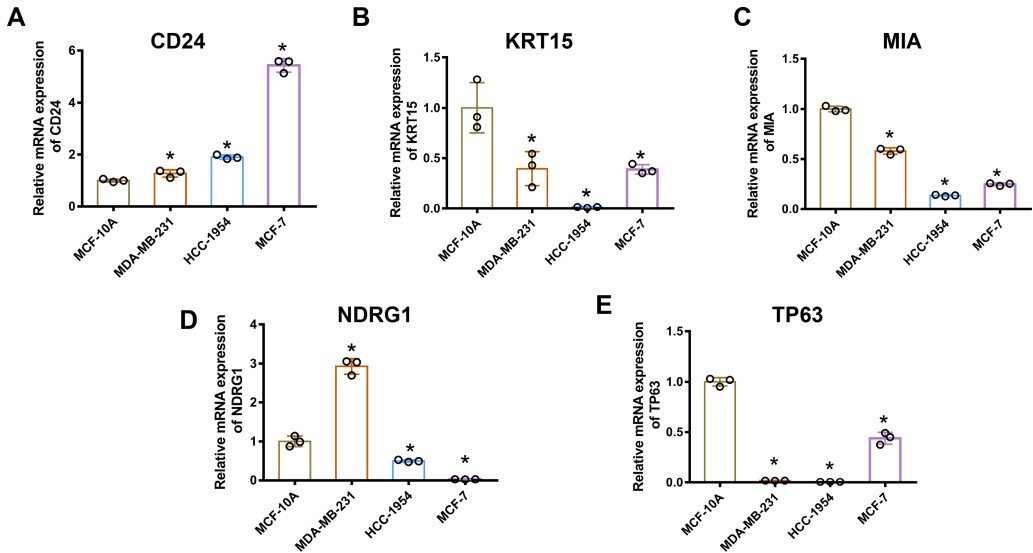

**Figure 6  MRNA levels of ARGs in MCF-10A, MDA-MB 231, HCC1954, and MCF-7 cell lines.** mRNA levels of CD24 (A), KRT15 (B), MIA (C), NDRG1 (D), and TP63 (E) determined by qPCR. The mean $\pm$ SD is shown. $^{*}$ $p < 0.05$. mRNA, messenger ribonucleic acid; ARGs, anoikis-related genes; MCF, Michigan Cancer Foundation; CD24, cluster of differentiation 24; KRT15, keratin 15; MIA, melanoma-inhibiting activity; NDRG1, N-Myc downstream regulated 1; TP63, tumour protein p63; qPCR, quantitative polymerase chain reaction; SD, standard deviation.

AGRs (CD24, KRT15, MIA, NDRG1, and TP63), which met the criteria, were identified as key genes and were used for univariate and multivariable Cox regression analysis combined with clinical characteristics of the TCGA samples. Furthermore, calibration curves demonstrated a good prognostic performance of the Cox regression model, with the most accurate prediction being made at the 3-year survival. DCA indicated the best clinical applicability in the 3-year survival. Precise prognosis prediction is essential for treating patients with breast cancer. This new information in our study might provide more information for developing prognostic tools. Therefore, the five ARGs might play significant roles in breast cancer progression and aggressiveness.

We confirmed the strong expression of CD24 in breast cancer through qPCR and IHC, which is consistent with previous reports (*Barkal et al., 2019*). CD24 is a mucin-like glycosylated glycophosphatidylinositol-anchored molecule, which could help resist anoikis in various tumours (*Li, Sun & Wang, 2015*). It has been suggested recently that CD24 is an immune checkpoint promoting immune evasion through its ability to bind to sialic acid-binding immunoglobulin-like lectin 10 (*Barkal et al., 2019*). Furthermore, it has been confirmed that CD24 might be a potential therapeutic target, according to preclinical studies (*Ni, Zhao & Wang, 2020*). Nevertheless, there are some conflicting reports on the role of CD24 in cancer (*Ni, Zhao & Wang, 2020*). Our study provided clear evidence of significant CD24 up-regulation at the protein and mRNA levels in breast cancer, and it has been suggested to be a high-risk factor resulting in poor prognosis in patients with breast cancer.

KRT15, a type I keratin, has been shown to be highly expressed in oesophageal carcinoma, colorectal cancer, and gastric cancer (*Chen & Miao, 2022*), whereas it is down-regulated in breast and prostate cancers (*Chong et al., 2012*). Nevertheless, its precise functional role and associated molecular mechanisms in breast cancer remain unclear. Our data suggests that KRT15 might be a tumour suppressor gene and could be associated with longer overall survival of patients with breast cancer. Furthermore, enrichment analysis proposed further avenues of exploration into the role of KRT15 in anoikis.

MIA is secreted by malignant melanoma cells and is associated with cell detachment from the ECM and the promotion of melanoma cell invasion and metastasis (*Schmidt & Bosserhoff, 2009*). To date, the relationship between MIA and breast cancer has not been reported. Our study is the first to reveal that MIA acts in a protective manner in patients with breast cancer. We also used qPCR to confirm that the MIA mRNA levels were lower in breast cancer cell lines compared with normal breast cells, which gave insights into breast cancer progression and metastasis.

Currently, the exact role of NDRG1in breast cancer remains controversial (*Joshi, Lakhani & Reed, 2022*). Herein, survival analysis demonstrated that NDRG1 might be an oncogene associated with a poor prognosis. Furthermore, IHC results from the HPA database demonstrated that NDRG1 protein expression was higher in breast cancer tissues. However, NDRG1 mRNA levels were increased in MDA-MB-231 cells and decreased in HCC 1954 cells and MCF-7 cells compared with normal mammary cells, indicating that NDRG1 might play distinct roles in different breast cancer subtypes. Consequently, we shall further examine the protein and mRNA levels of NDRG1 in additional clinical samples. Simultaneously, a study of the underlying molecular mechanisms of NDRG1 is required to determine its role in breast cancer.

Protein p63 encoded by the TP63 gene belongs to the p53 transcription factor family, and various isoforms are generated due to alternative splicing (*Gatti et al., 2019*). Two major subtypes of p63 (TAp63 and ΔNp63) have contrasting roles in breast cancer. TAp63 acts as a tumour suppressor, while ΔNp63 might promote tumourigenesis and progression in breast cancer (*Gatti et al., 2019*). Our study demonstrated that decreased TP63 expression is a negative and independent prognostic factor in breast cancer, and the TAp63 mRNA level was down-regulated in MDA-MB-231, HCC1954, and MCF-7 cells in laboratory tests, consistent with previous reports. In conclusion, five key ARGs were closely associated with breast cancer progression.

In this study, we initially examined the prognostic values and expression profiles of ARGs in patients with breast cancer. However, our study has some limitations. We did not perform a prognostic analysis of breast cancer subtypes, and the laboratory data did not comprise the key ARG protein levels in clinical samples and cell lines. We leave these questions to future studies and further investigate the molecular mechanisms of key ARGs in breast cancer.

# CONCLUSIONS

This study primarily aimed to determine the critical role of ARGs in breast cancer progression and search for valuable diagnostic and prognostic markers. The findings revealed that the ARG-based predictive model could accurately predict the survival of patients with breast cancer. We also investigated the possible BPs of ARGs and elucidated expression features and potential prognostic value for each ARG. Our data emphasize the significant roles of ARGs in breast cancer progression and provide novel research ideas for anoikis resistance.

# ACKNOWLEDGEMENTS

We are grateful to The Fang Zheng laboratory of Jinan University for instrument support and assistance.

## Funding

This work was supported by the National Natural Science Foundation of China (nos. 82074430, 81803979, and 81673979), the Guangdong Basic and Applied Basic Research Foundation, China (no. 2022A1515011674), the Natural Science Foundation of Guangdong Province, China (nos. 2018A030313393 and 2016A030313114), Science and Technology Program of Guangzhou, China (nos. 201803010051, 201707010245, and 201704020117) and the Fourth Batch of TCM Clinical Outstanding Talent Program of China (no. 444258). The funders had no role in study design, data collection and analysis, decision to publish, or preparation of the manuscript.

## Grant Disclosures

The following grant information was disclosed by the authors:
National Natural Science Foundation of China: 82074430, 81803979, 81673979.
Guangdong Basic and Applied Basic Research Foundation, China: 2022A1515011674.
Natural Science Foundation of Guangdong Province, China: 2018A030313393, 2016A030313114.
Science and Technology Program of Guangzhou, China: 201803010051, 201707010245, 201704020117.
Fourth Batch of TCM Clinical Outstanding Talent Program of China: 444258.

## Competing Interests

The authors declare there are no competing interests.

## Author Contributions

- Jingyu Cao conceived and designed the experiments, performed the experiments, analyzed the data, prepared figures and/or tables, authored or reviewed drafts of the article, and approved the final draft.

- Xinyi Ma conceived and designed the experiments, performed the experiments, analyzed the data, prepared figures and/or tables, authored or reviewed drafts of the article, and approved the final draft.
- Guijuan Zhang analyzed the data, prepared figures and/or tables, and approved the final draft.
- Shouyi Hong analyzed the data, prepared figures and/or tables, and approved the final draft.
- Ruirui Ma analyzed the data, prepared figures and/or tables, and approved the final draft.
- Yanqiu Wang analyzed the data, prepared figures and/or tables, and approved the final draft.
- Xianxin Yan conceived and designed the experiments, performed the experiments, prepared figures and/or tables, authored or reviewed drafts of the article, and approved the final draft.
- Min Ma conceived and designed the experiments, prepared figures and/or tables, authored or reviewed drafts of the article, and approved the final draft.

## Data Availability

The raw measurements are available in the Supplementary File and figshare: Jingyu, Cao (2023). Raw data and code. figshare. Dataset. https://doi.org/10.6084/m9.figshare.23938458.v1.

## Supplemental Information

Supplemental information for this article can be found online at http://dx.doi.org/10.7717/peerj.15475#supplemental-information.

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
