# Peer review of "Prognostic analyses of genes associated with anoikis in breast cancer"

_PeerJ, doi:10.7717/peerj.15475_

## Round 0.1 · original submission · Major Revisions

Dear Dr. Cao,

Thank you for your submission to PeerJ.

It is my opinion as the Academic Editor for your article - Integrative analyses of genes associated with anoikis in breast cancer - that it requires a number of Major Revisions based on reviewer comments.

We anticipate that you will revise the manuscript as per the comments in a point-by-point manner.

With kind regards,
Abhishek Tyagi
Academic Editor
PeerJ Life & Environment

Reviewer 1 ·

Basic reporting

In this manuscript Cao et al. studies the prognostic value of anoikis related genes (ARGs) in breast cancer and shows that 11 ARGs are differentially expressed in high and low risk groups that dictate patient survival. Their further analysis shows some 5 ARGs that are capable of predicting 3 year survival outcomes, but at the end, they finally emphasize on CD24 and NDRG1 that are overexpressed and lead to negative outcomes in BC. The manuscript is well written, offers novel insights into cancer anoikis research and is in fairly good shape for publication. However, following are my comments to further improve the quality of manuscript before publication.

Basic reporting:
Manuscript is written well and in professional manner. I have no major comments except following. I'd recommend that authors check the text for any typo or grammatical errors.

1. Fig. 1- I suggest elaborating more on this figure with more details (briefly describe what was done and why at each stage). In current form its not very descriptive and not very helpful to readers. At the very least, add full forms of the abbreviation used.
2. Fig. 3 D,E- display number of genes in both Up and Down regulated sections for ease of readers
3. Fig. 7 C-E- Figures seem cropped from the top. Suggest revisiting and providing new figures.
4. I suggest Fig. 2, 4 be moved to supplemental as they don’t contribute to the main message of the paper.
5. Methods: Please describe how the qPCR results (Fig. 9) were analyzed? Did authors use 2deltaCT method?
6. Typos or error in sentence phrasing at following places:
Line 304: ‘payed’ should be ‘paid’
Line 310: ‘future study’ should be ‘further study’
Line 318: ‘istantproteolysis’ need space in between
Line 357: ‘promots’ should be ‘promotes’
Line 359: ‘braest’ should be ‘breast’
Line 368 and 377: Please check sentence phrasing and grammar

Experimental design

1. Although all the analyses and bioinformatics approach in the study look reasonable, it is not clear at many points as to why authors choose to selectively use either TCGA or Geo BRCA datasets for their analyses OR what exact dataset has been used?

For instance, Fig. 3C- does this data combine both TCGA and BRCA datasets? Can they perform these analyses separately on both datasets how they have done for Figs. 3 D-I? Please clarify.

Why Fig. 5A only uses TCGA and not BRCA dataset? Have authors performed similar analysis on BRCA?

Similarly in Fig 6 (KM and ROC curves), what datasets are used and why? Please describe.

2. I appreciate authors following up their findings with validation using IHC and qPCR, however this is also one of my major concerns as in my opinion the data is highly limited and needs to be strengthened to some extent.

Fig. 8- All of these data are taken up from HPA and hence should be removed. If authors wish, they can reference this publicly available data in their manuscript. Authors should perform IHC staining of CD24, NDRG1 or other ARGs on BC patient tissues themselves, get it scored by pathologists and include that in the manuscript. Also having just one IHC sample is not meaningful and higher n would be necessary.

Fig. 9- There are more breast cancer cell lines (e.g. MCF7, MDA-MB-468, MX-1, ZR-75-1) authors could have assessed for qPCR analysis that authors could have evaluated to strengthen the data. This will help explain the discrepancy observed for genes like NDRG1 where they see opposite results for MDA-MB-231 and HCC-1954.

Fig. 9- I suggest authors also perform western blot or flow cytometry analyses to assess protein expression of CD24, KIRT15, MIA, NDRG1, and TP63 to validate their qPCR results.

3. Fig. 7- The description of the results (line 269-280) for this figure needs to be re-written. It is not clear (Fig. 7 C-H) on what basis authors determine that 5 ARG genes had best prediction for 3 year survival? What objective parameters were used to make this conclusion?

Validity of the findings

Conclusions made in the manuscript are overall supported by the authors' data.

In the conclusions, authors state in line # 392 that “Thus, it might contribute to clinical decision and treatment selection”. I think this line should be removed as authors’ data is still preliminary for any clinical implications yet.

Additional comments

No additional comments.

Reviewer 2 ·

Basic reporting

The manuscript needs extensive language editing in terms of spellchecks and also rephrasing in order to ensure the meaning of the work is unambiguously put forth. The subheading used should relate to the data rather than the names of the methods employed. Some of the subsections in the results section need to be clubbed in order to ensure coherence. Therefore, the manuscript needs to be restructured carefully. Figure 4C has not been cited in this manuscript. The discussion section should discuss the data presented in the manuscript.

Experimental design

The methodology employed can be represented as a schematic diagram for better understanding. The workflow chart in Figure 1 is inadequate and needs further refining. For example, Cox regression analysis results were not the ones used for IHC. Instead, IHC was done using a different set of samples. Furthermore, please explain the methodology in more detail. For example:
Line 114-115: Please explain “intersection” was based on what feature? Were these genes common in both databases, or was there another feature responsible for the same?
Flow cytometry and Western Blot experiments for some of these markers may be included to validate the data.
Fig 4d uses STRING database interactions. Please explain why the threshold was set at 0.15. Is this the same as "Required score" tab in the "Advanced settings in STRING database? If so, provide results with medium and high confidence scores in order to substantiate the claims made in this paper.
The reason why GSEA and GSVA were done is unclear and does not seem connected with anything substantial in the manuscript. The result regarding NOTCH and WNT pathways seems redundant as such information is already available in the literature.
The experimental design, in general, lacks coherence with the research problems and objectives stated in the beginning.

Validity of the findings

The manuscript seems to be at certain points indicating validating LASSO predictions which seems out of place. Please explain in detail what is constructed using LASSO and whether the output has been validated experimentally.
The “best prediction in 3-year survival” needs to be discussed in order to bring out the value of this manuscript.
Moreover, provide the reasoning for high CD24, with reference to the extant literature, which generally states CD24 has low expression in MDAMB231. Furthermore, state the ER/PR status of the IHC samples A and C, as the results might change with different breast cancer subtypes.
NDRG1 is not significantly differentially expressed based on the results shown in the figures. While Fig. 3h shows significance, Fig 3i clearly shows "n.s." written on NDRG1, which is in conflict with the statement in Line 227 "11 significantly differentially expressed ARGs....with same expression trends". Furthermore, NDRG1-related data needs further refining as a simple TCPA search shows different results in different breast cancer subtypes. Hence, the apparent "conflict" in your observation.
No connection between NOTCH and WNT signaling and the anoikis resistance or anoikis resistance genes is apparent from the data shown. The conclusions should have substantial data, as there were many other pathways that were enriched in the GSEA data shown.

Additional comments

The design of the experiments was not appropriate for breast cancer, as we now know that breast cancer cannot be looked upon as monolithic cancer. The study needs to be conducted with respect to different subtypes since it is widely known that the same treatment strategy might not work for a different subtype. Furthermore, different subtypes have different molecular signatures. Therefore, applying the inferences drawn in this paper across all subtypes might be erroneous. Therefore, it is recommended to differentiate the datasets first on the basis of the expression of ER, PR and HER.

---

## Round 0.2 · accepted · Accept

Dear Dr. Cao,
Thank you for your submission to PeerJ.
I am writing to inform you that your manuscript - Prognostic analyses of genes associated with anoikis in breast cancer - has been Accepted for publication.
Congratulations!

With kind regards,
Abhishek Tyagi


"In this study, a prognostic model of ARGs based on The Cancer Genome Atlas(TCGA) database using a least absolute shrinkage and selection operator analysis to evaluate the prognostic value of ARGs in BRCA." - this sentence has no verb

Reviewer 1 ·

Basic reporting

The revised manuscript is of professionally acceptable standards in terms of writing/English language.

Experimental design

No comment

Validity of the findings

No comment

Additional comments

The authors have satisfactorily addressed most of my concerns. The only remaining comment was the further protein/IHC validation of their findings, however, this should fall within the scope of future studies as pointed out by the authors in their response to reviewers.